# Methods of Ore Pretreatment for Comminution Energy Reduction

**Sefiu O. Adewuyi** *[ID], **Hussin A. M. Ahmed**[ID] and **Haitham M. A. Ahmed**[ID]

Mining Engineering Department, King Abdulaziz University Jeddah, Jeddah 21589, Saudi Arabia;
hussien135@gmail.com (H.A.M.A.); hmahmed@kau.edu.sa (H.M.A.A.)
* Correspondence: sefiuadewuyi@gmail.com; Tel.: +234-8070583557

**Abstract:** The comminution of ores consumes a high portion of energy. Therefore, different pretreatment methods of ores prior to their comminution are considered to reduce this energy. However, the results of pretreatment methods and their technological development are scattered in literature. Hence, this paper aims at collating the different ore pretreatment methods with their applications and results from published articles, conference proceedings, and verified reports. It was found that pretreatment methods include thermal (via oven, microwave, or radiofrequency), chemical additive, electric, magnetic, ultrasonic, and bio-milling. Results showed that the chemical pretreatment method has been used at an industrial scale since 1930, mainly in cement production. The microwave pretreatment results showed positive improvements at pilot scale mining applications in 2017. The results of ore pretreatment using electric and ultrasonic methods showed up to 24% and 66% improvement in energy consumption, respectively. The former and the latter have been piloted for gold and carbonate ore, respectively. Findings also showed that magnetic, radiofrequency, and bio-milling methods have not led to significant reductions in comminution energy. Based on energy reduction, safety, costs, stage of application, and downstream benefits, microwave and electrical pretreatment methods may be focused for applications in the mining industry.

**Keywords:** mining operation; ore milling; ore grinding; rock; liberation

## 1. Introduction

The mineral industry consumes significant energy from the national energy demand. The energy consumed in mining as a percentage of national energy consumption in countries such as Australia, USA, Canada, China, Saudi Arabia, and South Africa are 11 [1], 12 [2], 8 [3], 10 [4], 8 [5], and 17.4 [6], respectively. The sources of the energy (for the USA) and the cost trend are presented in Figure 1. In the mineral industry, comminution accounts for the largest share of energy consumption [7,8]. Comminution involves two-unit operations—crushing and grinding. Grinding is always a great concern because it accounts for about 50–70% of the total energy consumption [7,8]. Researchers have elaborately clarified why energy in grinding is so high. The main reason is that as material becomes smaller, more energy is required to create new surfaces. Unfortunately, most of the applied energy is usually lost as noise, heat, and electromechanical during the grinding operation; only about 1% is used for the materials' grinding [9–12]. Hence, there is a need to look for means of rationalizing energy consumption in the grinding process (Figure 2a) [12,13]. Comminution energy depends on minerals to be comminuted as per their type, amount, and properties. Moreover, it relies on the used machine and its operation parameters and skills of operating manpower [14]. The trend of comminution energy (Canada mining industry) between 1990 and 2015 is as presented in Figure 2b. The mean and standard deviation of the energy consumed within these years are 70.71 petajoules and

7.14, respectively. The comminution energy increased by 6.97 petajoules within the period, with a possible increase in future due to declined ore grades and the increase in demand for minerals [14].

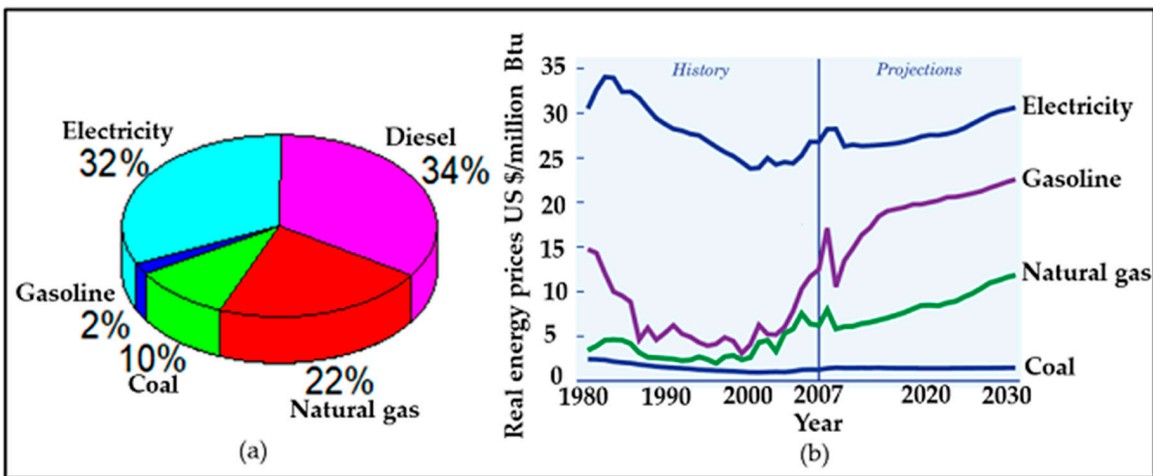

**Figure 1.** (**a**) Mining operations energy sources (data sources: [13,15]). (**b**) Energy costs trend [16].

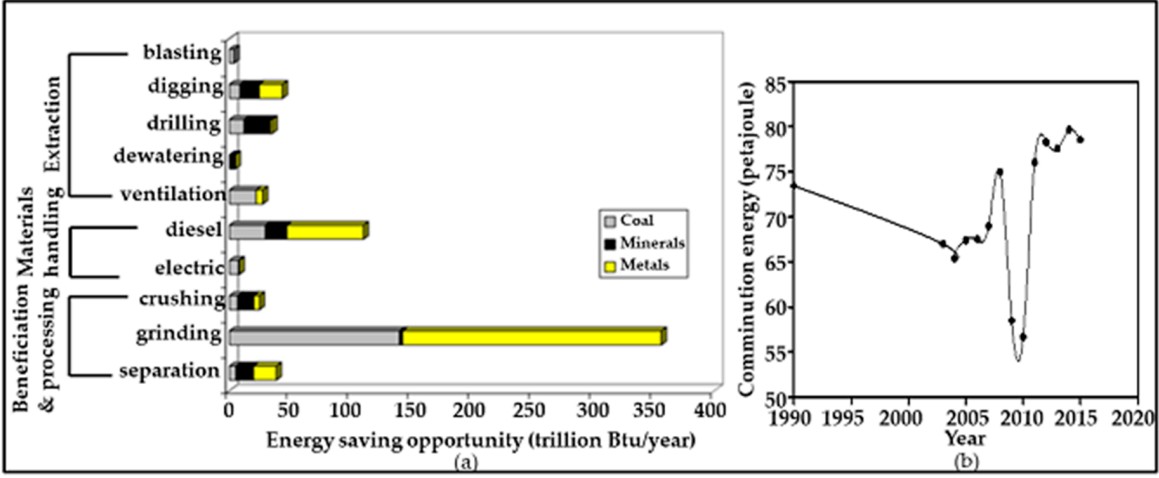

**Figure 2.** (**a**) Energy-saving opportunity for energy intensive process in mining [13]. (**b**) Comminution energy trend (Canada) between 1990 and 2015 (calculated based on 50% of total mining energy) (data source: [17]).

To ameliorate the situation, early studies have been tailored towards the optimization of grinding circuits. Some of the approaches are the optimization of the mill load, filling ratio [18], ore to media ratio, media size distribution [19], mill length to diameter ratio, and adding mill riffles [20]. In the last few decades, the pretreatment of ore prior to comminution has been proposed [21]. Pretreatment denotes an independent operation performed on ores prior to grinding. The main objective of pretreating ore is to create intergranular cracks so that the grinding operation can become easier, which can lead to a reduction in the grinding energy [22–26]. Among the pretreatment methods are thermal (via furnace, microwave, or radio frequency), chemical additives, electric, magnetic, ultrasonic, radio frequency, and bio-milling. These methods can be categorized as presented in Figure 3. Some of the pretreatment methods have showed positive results towards the reduction in ore's grinding energy requirement but are yet to be commercialized. Previously, some review works had been performed that focused on a single method—chemical additive [27], thermal via furnace [28], thermal via microwave [29], and electric pretreatments [30]. Somani et al. (2017) provided a combined review of thermal, ultrasonic, and electrical pretreatments [21]. In this work, a holistic review of well-known pretreatment methods

is attempted in order to give a concise overview of the development in comminution energy reduction and provide for future research needs.

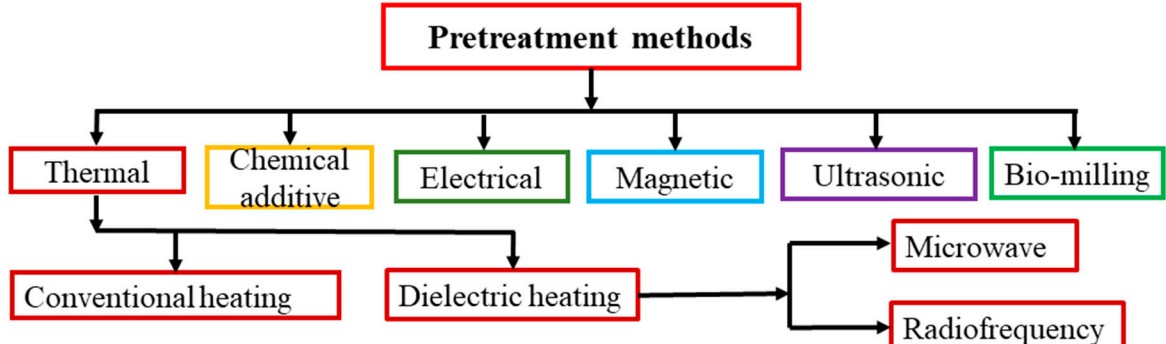

**Figure 3.** Classification of pretreatment methods.

## 2. Thermal Pretreatment

Thermal pretreatment is a method that applies heat energy through conduction, convection, or radiation to create granular or inter-granular cracks in rocks [21]. When the temperature of the material changes, the physical properties of the material are altered, which leads to the displacement of some grains leading to fractures. Basically, there are two means of applying heat energy to rocks—through conventional (furnace) and electromagnetic (radiofrequency and microwave) or dielectric heating.

### 2.1. Thermal Pretreatment via Furnace

The use of a furnace to heat rock is usually referred to as conventional heating. When rock is subjected to heating, stresses are developed within the rock matrix that leads to cracking as a result of thermal expansion and contraction. Thermal pretreatment was introduced in mineral comminution with the prime objective to increase liberation and reduce the energy used in the process [31]. Since rock is an aggregate of minerals, the heating of rocks causes crystals to expand in different orientations depending on the mineral constituents. The expansion of crystals may cause the internal cracking of rocks, which can reduce the competence of rocks before comminution. Inter-crystalline may open up around 200 °C upward, which can increase the existing discontinuity and create new ones in granite [32], gypsum, and celestite [33]; however, this may not be the case for all rock types because ore texture, crystallinity, and size has a significant effect on ore response to thermal pretreatment [34,35]. In addition, a crystal shape may contribute to the crack pattern when grains are displaced due to the expansion of the minerals. The extent of the cracking of rock in different directions depends on the thermal confined stress at any direction. This approach has been used in different studies to create cracks in rocks since early work in the 20th century [36,37]. Omran et al. (2015) demonstrated that, as the temperature of the furnace increases (400 °C and above, at 1 h residence time), the cracks developing in the iron ore matrix increase (approximately 10% increase for every 100 °C) and consequently, the particles' liberation is improved (Figure 4) [34]. However, the maximum temperature after which increasing the temperature has no effect on the particles' liberation from the iron ore still needs to be established. The thermal pretreatment of rock is of interest in rock drilling and excavation, underground storage, nuclear waste, deep petroleum boring, tunneling, dam, reservoir [38], geothermal [39], archeological study [40], building construction [41], and ore comminution and beneficiation [42].

Effect of Thermal Pretreatment (via Furnace) on Ore Grindability

A review of early work on conventional heating as related to rock grindability was discussed by Fitzgibbon and Veasey (1990) [28]. It was reported that the grinding resistance (grindability) of rock can be lowered using a thermal pretreatment approach. Thermal alteration in rocks has different

effects, such as structural damage, phase-change, decomposition, and desorption [43]. The anisotropic nature of minerals causes thermal stress concentrations at points and grain boundaries, which may lead to the fracturing of individual mineral grains when heat is applied to rocks [43].

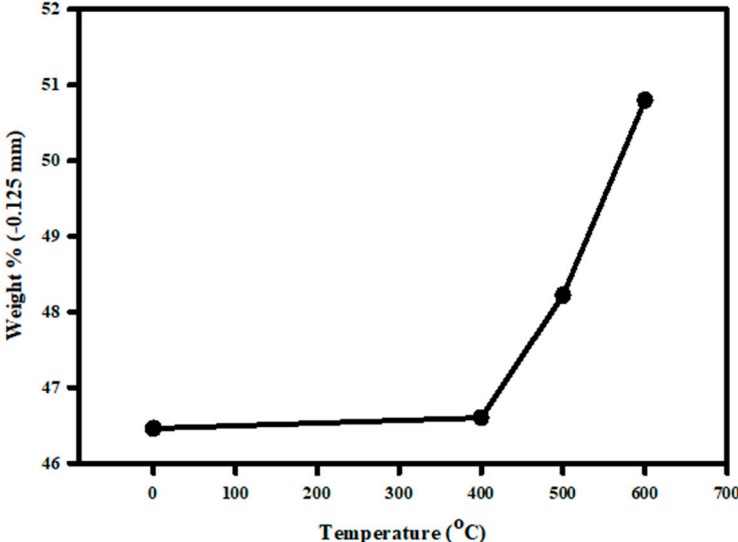

**Figure 4.** Effect of thermal treatment via furnace on iron ore grinding (100 g, 2.45 GHz, and 1 h) (data source: [34]).

The effect of thermal pretreatment on the grindability of celestite and gypsum was studied [33]. The rock sample of size fractions −1.168 + 0.6 mm (500 g) was ground using a ceramic ball (20 and 25 mm diameters, 588 g). The grinding time ranged between 5 and 30 min at 5 min intervals. The work indexes of untreated celestite and gypsum calculated using the Hardgrove method were 6.76 kWh/ton and 5.18 kWh/ton, respectively [33]. It was reported that there was no significant effect of heat pretreatment on celestite's grindability within 200 °C, while that of gypsum decreased by 22.8% within the same temperature range. This result was expected because gypsum, being one of the hydrate minerals, can decompose easily due to the removal of water molecules when exposed to temperatures within 10–250 °C. When dehydration occurs, gypsum ($CaSO_4 \cdot 2H_2O$) may transformed to plaster of Paris (hemihydrate mineral: $CaSO_4 \cdot O \cdot 5H_2O$), which usually leads to structural failure [33,44]. This shows that the existence of water molecules in the rock structure may help ore response to thermal treatment. Conversely, celestite required higher temperatures up to 1140 °C before effective changes could be observed [33,45].

Recently, the impact of the thermal pretreatment of manganese ore (selected from Qom mining site, Iran) was investigated [44]. Different size fractions (−1.7 + 1.18 mm, −1.18 + 0.6 mm, −0.6 + 0.3 mm, and −0.3 + 0.15 mm) were separately investigated to determine the appropriate size range in which thermal pretreatment can cause a significant effect on the breakage characteristic of manganese ore. A sample was thermally treated in a furnace for 60 min at 750 °C. The treated sample was ground in a ball mill (diameter; 20 cm, height; 25 cm) using ball charges of different sizes of 8.869 kg. An untreated sample of the same mass was ground under the same grinding conditions and the results were compared. The specific rate of breakage approach was adopted in the study, using the first order kinetic model. The slope of the semi-logarithm curve for mass retained against grinding time was used to estimate the breakage characteristic of the ore. The results of the breakage characteristics of treated (thermal treatment) and non-treated manganese ore were compared. An improvement in grinding rate of 37% was obtained for the size fraction −0.30 + 0.15 mm. This result may be attributed to the fact that heat better penetrated to the lower size range than the higher one when treated under the same conditions. The results from different studies related to the pretreatment of ore via furnace are presented in Table 1.

**Table 1.** Summary of thermal pretreatment via furnace on the grindability of minerals/ores.

| Mineral/Ore | Mass (kg) | Size Fraction (mm) | Treatment Temperature (°C) | Soaking Time (min) | Cooling | Improvement in Grindability | Reference |
|---|---|---|---|---|---|---|---|
| Celestite | 0.5 | −1.168 + 0.6 | 200 | 60 | Air | 0 | [33] |
| Cassiterite | - | - | 650 | - | Water | 45 | [46] |
| Gypsum | 0.5 | −1.168 + 0.6 | 200 | 60 | Air | 22.8 | [33] |
| Manganese | | −0.3 + 0.15 | 750 | 60 | Air | 37 | [42] |
| Quartzite | 0.65 | −4 + 0.25 | 650 | 65 | Water | 18 | [24] |
| Quartzite | 0.65 | −4 + 0.25 | 650 | 65 | Alkali | 32 | [24] |
| Quartzite | 0.65 | −4 + 0.25 | 650 | 65 | Acid | 28 | [24] |
| Quartzite | 0.65 | −4 + 0.25 | 650 | 65 | Salt solution | 20 | [24] |
| Hematite | 0.1 | - | 600 | 60 | - | 4.2 | [34] |

Downstream Benefits, Economic Assessment and Industrial Applications of Thermal Pretreatment (via Furnace)

Despite improvements in grindability that may reach up to 45% for some ores (Table 1), thermal pretreatment via furnace has neither been piloted nor adopted in the mining industry. The following challenges have been associated with the method: 1) non-uniformity in rock heating; 2) surface heating; 3) not environmentally friendly—it releases gases to the environment; 4) safety issues related to high temperature; and 5) high energy consumption. Despite all these challenges, thermal pretreatment through furnace is still being pursued, not only to improve the grindability of ores but also to improve downstream operations. Dash et al. (2019) demonstrated that thermal treatment can improve the magnetic separation of low-grade hematite ore [47]. The representative samples (200 g, 10 mm) were treated in a laboratory furnace and the samples were water quenched after the treatment times were reached. It was then ground to −75 µm using a ball mill. The analysis of samples after magnetic separation (wet high intensity magnetic separation (WHIMS); solid % = 25) showed that the iron yield can be improved within a range of 15–20% when hematite is treated between 500 and 800 °C [47]. Early studies show that the thermal treatment of tin ore can save 93 W/t of the ore processed, however 117 kWh/t will be consumed by the furnace at 100% efficiency [48]. This shows that the method may not be economically viable. The issue of the high energy consumption of the furnace is the major challenge; however, further investigation shows that an improvement in liberation usually leads to a high recovery, which may offset the energy consumed during the pretreatment. In fact, a 1% increase in recovery has been argued to be enough to cover the expended energy on the pretreatment via furnace [28]. Even when the improvement in liberation is insignificant, thermal pretreatment can still lead to a better recovery, especially for iron ores due to increases in their magnetic properties [47]. Nevertheless, further research is still needed to investigate the downstream benefits of the thermal pretreatment (via furnace) which will lead to thorough economic assessments and possible applications in the mining industry.

*2.2. Thermal Pretreatment via Electromagnetic Waves (Dielectric Heating)*

The transfer of energy without the need of a specific material medium of propagation is termed electromagnetic (EM) waves. EM waves are categorized based on wavelength ($\lambda$) and frequency ($f$) (Figure 5), since all EM waves travel at the same speed ($c = f\lambda$).

Among the EM spectrum, radio waves and microwaves (MW) are being used for heating applications in some industries, such as food, geological exploration, pharmaceutical, plastic, construction, medical therapy, and mining. Radio wave heating is usually referred to as radiofrequency (RF) heating [49]. Both RF and MW techniques are referred to as dielectric heating because their heating mechanism relies on the dielectric properties of the material to be heated. Materials with significant dielectric properties respond to dielectric heating. In mining, dielectric heating has gained the attention of researchers because of its peculiar advantages over conventional heating, such as the production of clean energy, fast processing, heating uniformity, and better heat penetration to the ore matrix.

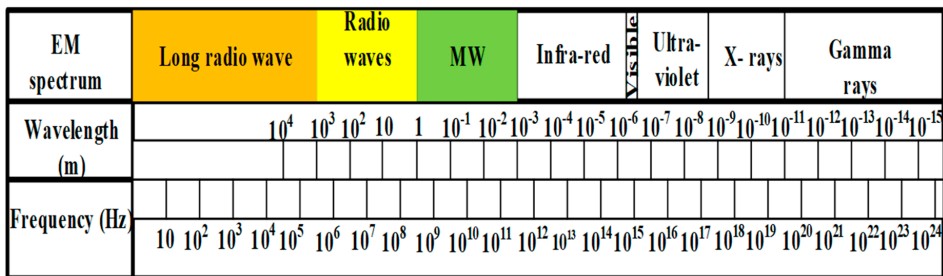

**Figure 5.** Electromagnetic spectrum.

In contrast to heating through furnaces, which employ heat transferred through convection, conduction, and radiation from the material's surface, dielectric heating involves energy conversion from electromagnetic into heat energy. When material is subjected to an electromagnetic field, electric and magnetic polarization may arise within the frequency of the electromagnetic device used in the process, which may lead to heating. This heating occurs in different mechanisms: dipolar polarization, conduction, and interfacial polarization. In dipolar polarization, the atomic dipole reorientation occurs, but there exists a phase difference between the dipoles and the electric field orientation, which leads to the heating of the material. When a material consisting of a significant electrical conductor is subjected to an electromagnetic field, ions or electrons move which leads to electric polarization. This causes the heating of the material due to its electrical resistance. In such cases, the heating mechanism is through conduction (like that of conventional heating), and a high microwave power and long residence time will be required to make a significant effect on the rock's strength reduction. In rocks with heating behavior such as Goethite (hydrated iron ore), the iron mineral heats, but the bulk ore must be heated to a point at which the hydroxy ion of the water molecules is released. For interfacial polarization, a sample consisting of conducting and non-conducting material with a significant dielectric property is subjected to an electromagnetic field such that both dipolar polarization and the conduction heating mechanism occur. Therefore, when ore is subjected to an electromagnetic filed, both the atomic dipole and electric polarization may occur, leading to microcracks. The extent of the microcracks depend on whether there is an existing fracture and the arrangement of crystals and even the crystal shape, which may help to increase the intergranular cracking.

The main difference between the microwave and radiofrequency means of dielectric heating is their frequency range, which determines the device's configurations. Microwave radiation has a higher frequency compared to the radiofrequency (Figure 5). However, within the frequency range of radio waves, 10–100 MHz is mostly employed for a heating purpose [49]. The typical device set up for the two approaches is as presented in Figure 6. Both methods have been explored for the dielectric heating of ore with the sole aim to reduce its strength and consequently, the comminution energy can be minimized.

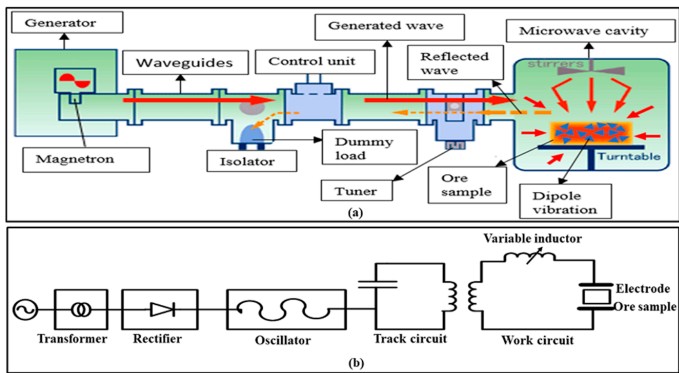

**Figure 6.** (**a**) Ore pretreatment in a microwave oven (modified after [50]). (**b**) Ore pretreatment in a radiofrequency device [51].

### 2.2.1. Effect of Microwave Pretreatment on Ore Grindability

Research towards the microwave treatment of minerals was firstly performed in 1975 [52]. The research basically focused on the minerals' responses to microwave treatment, with an emphasis on their dielectric properties. Findings suggested that the approach can be used to heat the ore with minimal processing time compared to conventional heating. It was later demonstrated that ores have varying degrees of response to microwave treatment—hyperactive, active, and difficult to heat [22]. Magnetite, pyrrhotite, and chalcopyrite were grouped as hyperactive minerals because they responded well to the microwave radiation treatment. Chalcocite, galena, and pyrite were active, while albite, marble, quartz, and other gangue minerals were difficult to heat (Figure 7; difficult to heat minerals were arranged downward in decreasing order of their response to the microwave treatment) [53].

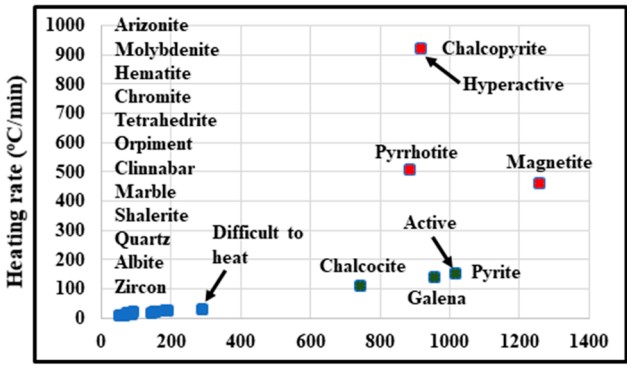

**Figure 7.** Heating response of minerals in a microwave (data source: [53]).

Afterwards, an attempt was made to investigate the microwave pretreatment's effect on iron ore grindability [7]. The next notable research focused on the effect of variation in the mineralogy of ores and the subsequent impact on their grindability when pretreated in the microwave [54]. Since then, many studies in this direction have been conducted with a focus on ore size [55], grain size [56], texture [57], microwave parameters (exposure time and microwave power), and the mode of cooling system after the microwave treatment [54]. The influence of microwave radiation on the material is usually more pronounced on larger particle sizes than on fine particles [58]. Coarser particles exhibited more fractures than fine particles when the same microwave power and residence time were applied [34,35,54,59]. Omran et al. (2014) demonstrated that iron ore with a larger grain size responded better to microwave radiation than a lower one treated under the same microwave conditions (Figure 8) [55,56].

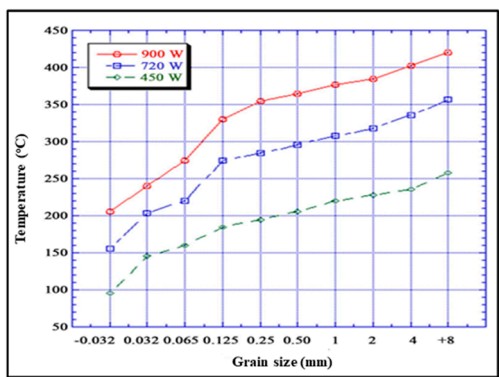

**Figure 8.** Effect of the grain size and microwave power on the iron ore temperature after the microwave (2.45 GHz) treatment (reproduced with permission: [55]).

As the size of the ore particles becomes smaller, the material hardness and resistance to grinding usually increases, which reduces the probability of creating flaws in the particles. Hence, a higher energy will be required to grind such particles [60]. However, when particles are pretreated in the microwave, the mineral liberation may reach the desired size quickly. To achieve that, the particle size after which microwave treatment has no improvement on the rock strength's reduction must be established, which is limited in the literature; hence, there is a need for further studies [58]. However, a simulation study on coal suggested that at the diameter 50 mm and height within 60 to 100 mm, coal's response to microwave radiation (2.45 GHz) was at optimum conditions [58]. The findings also indicated that the higher the height of the sample, the lower the temperature, but the better the electric field and temperature distribution [58]. Furthermore, at a large size, the hardness of the rock could be reduced when exposed to microwave radiation. This was demonstrated by Sikong and Bunsin (2009) using granite samples selected from Thailand [61]. The prepared representative samples ($16 \times 16 \times 30$ mm$^3$) were labeled thus: the dry sample, air cooled after the microwave treatment, was D-D; D-W was the dry sample water quenched after the microwave treatment; W-D was the wet sample (water soaking for 60 min) air cooled after the microwave treatment; and W-W was the wet sample (water soaking for 60 min) water quenched after the microwave treatment. The hardness of the samples before and after the microwave treatment (2.45 GHz, 600 W) was determined, and the results are presented in Figure 9a. The authors concluded that the microwave treatment of granite samples at lower exposure times has a beneficial effect on the reduction in rock hardness, and that soaking the granite samples in water hampered the reduction in rock hardness.

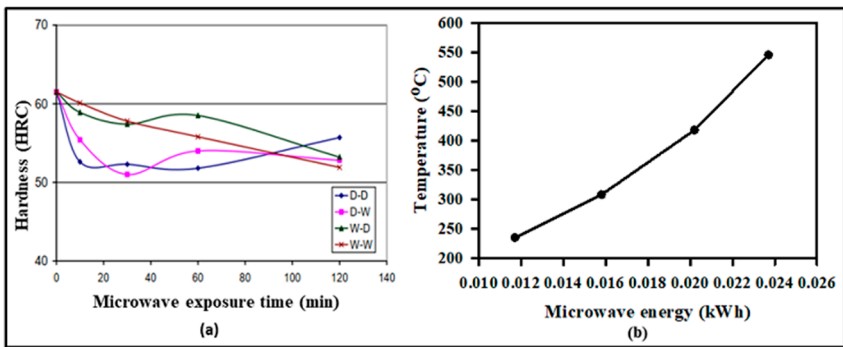

**Figure 9.** (**a**) Effect of the microwave exposure time on the hardness of granite (reproduced with permission) [61]. (**b**) Response of high phosphorus oolitic iron ore to microwave energy (data source: [34]).

For the influence of ore texture on the ore's response to microwave pretreatment, Batchelor et al. (2015) used lead–zinc, nickel, and copper ores to demonstrate that a high reduction in strength can be achieved in ores with a consistent texture after exposure to microwave radiation [57]. Microwave applied energy has a major influence on ore response to microwave radiation. This was studied using a high phosphorus oolitic iron (HPOI) ore selected from Egypt [34]. In the study, a 100 g representative sample was subjected to microwave radiation ($f$ = 2.45 GHz), and the final bulk sample's temperature and the microwave's energy consumption were measured using a thermocouple and energy meter (CLM 1000), respectively. The results indicated that the higher the applied energy, the better the final temperature reached by the samples (Figure 9b). A scanning electron microscopy (SEM) analysis of untreated and microwave-treated samples indicated that the higher the applied energy, the better the intergranular cracking in the microwave-treated ore. At a microwave (900 W) exposure time of 80 s, transgranular cracking occurred within the oolite, and part of the sample melted at 90 s exposure time. It was concluded that microwave pretreatment can improve the liberation of oolite from other minerals in the ore; consequently, grinding energy can be saved [34].

Omran et al. (2015) compared microwave (900 W, 60 s) and conventional heating (furnace—600 °C, 1 h) pretreatments using HPOI ore. The same grinding operation was performed for the untreated

and pretreated samples. The results showed that under size products (0.125 mm) increased—from 46.6% to 59.76% and 50.80%, and the equivalent to approximately 80% and 30% intergranual cracks developed in the samples for the microwave and furnace-treated samples, respectively. Under the treatment conditions, the energy consumption for the microwave was 0.0237 kWh, while that of the furnace was 5.33 kWh [34]. This suggests that the furnace consumed about 224.9 times the energy of the microwave, indicating that the microwave may be more economical than the furnace as an ore pretreatment method. Some of the studies tailored towards energy reduction in comminution, using the microwave pretreatment method as presented in Table 2.

**Table 2.** Summary of the thermal pretreatment via microwave (MW) (frequency = 2.45 GHz) on the grindability of minerals.

| Ore/Mineral | Mass (kg) | Size Fraction (mm) | MW Power (Kw) | Time (min) | Improvement in Grindability (%) | Reference |
|---|---|---|---|---|---|---|
| Magnetite | 0.35 | −3.36 | 3.0 | 3.5 | 21.4 | [7] |
| Hematite | 0.35 | −3.36 | 3.0 | 3.5 | 23.7 | [7] |
| Tactonite | 0.35 | −3.36 | 3.0 | 3.5 | 18.2 | [7] |
| Carbonatite * | 0.50 | −22.5 | 2.6 | 1.5 | 85.0 | [54] |
| Ilmenite * | 0.50 | −22.5 | 2.6 | 1.5 | 92.0 | [54] |
| Gold * | 0.50 | −22.5 | 2.6 | 4 | 0 | [54] |
| Copper * | 0.50 | −22.5 | 2.6 | 1.5 | 68.0 | [54] |
| Copper-zinc | 0.50 | - | 2.6 | 1.5 | 50.0 | [62] |
| Copper-zinc * | 0.50 | - | - | 1.5 | 65.0 | [62] |
| Copper * | 0.50 | - | 2.6 | 1.5 | 70.0 | [62] |
| Lead-zinc * | 3.00 | −19 + 2 | 4.0 | 5.0 | 30.0 | [63] |
| Ultramafic nickel | 0.10 | −1 + 0.425 | 1.2 | 15 | 3.6 | [64] |
| Iron | 0.50 | −19.05 + 12.7 | 0.9 | 2.0 | 50.0 | [65] |
| Lignite | 0.10 | −4.45 + 0.154 | 0.9 | 0.5 | 81.0 | [66] |

* Water cooling after microwave treatment.

Downstream Benefits, Economic Assessment and Industrial Applications of Microwave Pretreatment

Microwave pretreatment has been employed by many researchers at laboratory scales, as earlier discussed. Since 1991, when the first laboratory study of the method was performed with promising results for reducing comminution energy [7], the main issues are at the technological level of finding a microwave oven suitable for large-scale ore treatment and the economic feasibility of the method. There is no divergence of opinion that the improvement in grindability alone could not be used to adjudge the economic viability of the microwave pretreatment method. Walkiewicz et al. (1991) discussed that other benefits that can make microwave pretreatment economically viable include the reduction in the tear and wear of the mill, the mill liner, and the milling medium; and a possible increase in throughput with the reduction in recycled ore. Apart from these, the method can increase the grade and recovery of targeted minerals or elements of interest in some rocks [67]. This had been studied using Ilmenite ore [68]. Untreated and microwave-treated (at different power levels—1.3 kW and 2.6 kW) representative samples (200 g, −16 mm) were crushed to 100% passing 220 μm. A two-stage high-intensity wet magnetic (first, 0.045 T was used to remove magnetite; second, 1 T was used to remove ilmenite) separation of Titanium (Ti) from the ore was performed for each of the samples. The result showed that the grade of Ti increased from approximately 1.8% to 3.5%, equivalent to a 7.2% increase in recovery for the microwave-treated samples at 2.6 kW. Indeed, for samples treated at 1.3 kW, the Ti grade increased from approximately 1.8% to 4.4%, equivalent to a 12.8% improvement in recovery [68]. The heap leaching of fine disseminated minerals usually consumes time and leads to a low recovery [69,70]. Therefore, the pretreatment of ore before heap leaching was suggested to improve the process [69]. This was demonstrated using sulphide ore [69]. The results showed that the microwave treatment ($f$ = 2Hz, pulse time = 100 μs, power = 5.6 kW, time = 30 s) of the samples (+9.2 −12.5 mm, 6 kg) prior to heap leaching (800 mL solution; 14 g/L sulphuric acid +3.75 g/L ferric sulphate, at 25 °C) improved the yield in the range of 7% to 12% [69]. Cai et al. (2018) studied the combined effects of the microwave pretreatment, acid leaching, and magnetic separation of high phosphorus oolitic hematite (HPOH) [71]. The first phase of the study indicated that the microwave pretreatment (2.5 kW, 45 s, water quenched after treatment) of the representative HPOH samples reduced their work index from 15.25 kWh/t to 10.11 kWh/t. The next phase was used to study the effect

of microwave pretreatment on magnetic separation, while in the last stage, the combined effect of microwave pretreatment and acid leaching (concentrated hydrochloric acid, 1:1; solid:liquid, 45 min) on the magnetic separation (magnetic intensity = 900 kA/m, pulse frequency = 45 MHz) was investigated. In all cases, the improvements in hematite liberation, iron grade, recovery, and dephosphorization were analyzed, and the results were as presented in Table 3.

**Table 3.** Improvement in hematite liberation, iron grade, recovery, and dephosphorization [71].

| Method | Liberation (−0.038) (%) | Grade (%) | Recovery (%) | Dephosphorization (%) |
|---|---|---|---|---|
| Microwave | 30.11 | 5.65 | 17.99 | 3.27 |
| Microwave + Acid leaching | 54.80 | 14.26 | 34.62 | 43.49 |

The effect of microwave pretreatment on the downstream process has been studied on the bioleaching of massive zinc sulphide ore [70]. Different coarse particle sizes were microwave pretreated (frequency = 2.45 GHz, time = 1 s, power = 5.50–5.92 kW), and a continuous column (10 L, 140 mm diameter, and 500 mm height) leaching operation was performed for 350 days. The mic-organism in the leaching process was L. ferriphilum, which acts as a catalysis in the oxidation of Fe (II) to Fe (III). Bioleaching of the same particle size and similar ore material using the same approach has been conducted earlier [72]. The findings showed that the microwave pretreatment of the ore improved the efficiency of the bioleaching operation, as presented in Table 4.

Microwave pretreatment indeed has downstream benefits that can make it economically viable [73]. Most of its downstream benefits are related to the increased grade and recovery, especially for fine mineral particles of interest disseminated in gangue. Extracting such mineral through microwave-assisted leaching has been shown to improve the mineral grade and recovery significantly. The detail of some of the early work using this approach can be found in the literature [73].

**Table 4.** Zinc leaching efficiency for non-treated and microwave-treated samples [70].

| Size Fraction (mm) | Zinc Leaching Efficiency (%) | | |
|---|---|---|---|
| | Non-Treated [74] | Non-Treated [72] | Microwave Treated [72] |
| −5 + 4.475 | 79.4 | 73.7 | 93.1 |
| −16 +9.5 | 68.7 | 65.6 | 81.3 |
| −25 + 19 | 59.1 | 58.7 | 72.0 |

An economic analysis of the microwave processing of arsenopyrite gold ore (200 t/day) was performed by EMR Microwave Technology Corp. (Fredericton, NB, Canada) in 1997. The results of their findings suggested that microwave processing of the ore was economically viable in both capital and operating costs [73]. The obtained results encouraged the EMR Microwave Technology Corp. (Canada) to conduct a pilot scale study using refractory gold ore, which was probably the first of its kind [67]. In the study, a fluidized bed reactor was developed and coupled with a microwave to produce gold concentrate. The use of the developed technique caused the conversion of pyrite to hematite and elemental sulfur, which led to gold liberation from the ore matrix. The economic analysis of the process as compared to the pressure oxidation and roasting methods is presented in Table 5.

**Table 5.** Economic comparison of microwave reactor with pressure oxidation and roasting method for the treatment of refractory gold (200 t/day concentrate from 2000 t/day operation) [67].

| Method | Operating Costs (US$/t) | Capital Costs (US$ Million) |
|---|---|---|
| Pressure oxidation | 33.58 | 26.50 |
| Roasting | 13.74 | 6.90 |
| Microwave reactor | 8.60 | 3.84 |

The microwave-assisted floatation of copper carbonate ore has also been suggested to be economically viable [74,75]. Comparative batch floatation tests were conducted on untreated and microwave-treated ore (5–12 kW, 0.1–0.5 s), and the results suggested an improvement of 6–15% copper recovery. The best scenario of the economic study of the process indicated a less than two years payback period for the microwave-assisted floatation method [74,75]. The parameters to scale up the technology were also suggested. It was concluded that with a microwave power density of approximately $10^9$ Wm$^{-3}$, and a microwave cavity capable of treating 100–1000 t/h of ore at approximately 0.1 residence time, the process would be commercially viable [74,75].

Recently, a pilot scale study of the microwave pretreatment method has also been conducted, with a possible reduction in comminution energy of up to 9%. It was demonstrated that, apart from the improvement in grindability, the method has the advantages of improving recovery, reducing the wear and tear of ball charges, and increasing the life span of the comminution equipment [25]. Despite the promising results of the microwave pretreatment method, it has not been adopted either in the cement nor the mining industry. The challenges that impair progress in the adoption of the microwave pretreatment in mining and cement industries are the needs for more understanding of microwave interaction with materials, multi-disciplinary research, expertise in microwave engineering, and more pilot scale demonstration of the method for the most demanding ores [76]. Additionally, a design of a high-power large-scale industrial microwave oven is highly demanded. A microwave model study suggests that to achieve the fast processing of ore within 0.002 to 0.2 s, a power density of $1 \times 10^{10}$ W/m$^3$ to $1 \times 10^{12}$ W/m$^3$ is required [77]. Despite the amount of expended efforts by researchers to demonstrate the applicability of the microwave pretreatment method in reducing the comminution energy, a study has not been conducted to envisage the overall energy that can be saved by considering the full cycle of mineral production. This could be as a result of limited information as to the estimation of the mineral production cycle in terms of electrical energy consumption. A report by the Mining Association of Canada and Natural Resources could be a good source of guidance for estimating the total energy saved in mining activities when microwave pretreatment is incorporated into the mineral production cycle [78].

### 2.2.2. Effect of Radiofrequency Pretreatment on Ore Grindability

Radio frequency dielectric heating has not drawn the attention of researchers like the microwave has, even though significant research had been conducted for its application in the food industry. Little is known about the possibility of the method to improve the grindability of ore. The first attempt toward this objective was performed using dolerite, marble, and sandstone [51]. The findings indicated that the product size distribution of RF-treated samples improved compared to the as-received samples for the dolerite and sandstone, which may suggest an improved grindability, while that of marble remained the same. Nevertheless, larger particle sizes increased for the studied samples, indicating that the RF method is not appropriate to reduce the comminution energy of the studied samples. There is a need to investigate this technique using other ores [79].

## 3. Chemical Additive Pretreatment

Chemical additives pretreatment is one of the oldest and probably the most convenient means of pretreating ore before grinding. Generally, there are two types of chemicals used, mostly for cement production. The first type is used to modify the surface structure of the particle to improve grinding, while the second type is used as a cement strength enhancer [80]. Any chemical material that can cause an improvement in ore size reduction when mixed with the feed is termed a chemical additive. It is usually referred to as a grinding aid/additive, since it is applicable to ore at the grinding stage. The grinding aid can be applied in both dry and wet grinding operations. Dry grinding, unlike the wet one, does not require the addition of liquid during the grinding operation. Dry grinding is mostly performed in cement production, while both dry and wet grinding are applicable in other mineral production. Grinding aid is usually active on the ore surface. It reacts with ore molecules to cause

local stresses and allows fragmentation at the grain boundaries during grinding. This may cause an improvement in the size distribution and reduce the agglomeration of particles [27].

*Effect of Grinding Aids on Ore Grindability*

Considerably more research has been carried out on the use of grinding aids to improve the grindability of material in dry grinding operations than in wet grinding. This has occurred as a result of the amount of cement clinker needed to be ground daily in tumbling mills. The grinding of cement clinker to the desired fine size is one the greatest concerns in the cement industry and accounts for huge operation costs in term of energy consumption (Figure 10a). The use of grinding aids in the cement industry to improve the grindability and product throughput of the cement clinker dates to 1930 [81]. Most grinding aids are organic liquids, such as tri-ethanol amine, glycerol, alcohols, propylene glycol, organosilicones, diethylene glycol, and resins, etc. [27,81]. Grinding aids are added to the clinker during the final grinding stage (Figure 10b) to reduce the comminution energy. The summary of the findings from the literature on the use of some grinding additives for the improvement of cement clinker's grindability is as presented in Table 6.

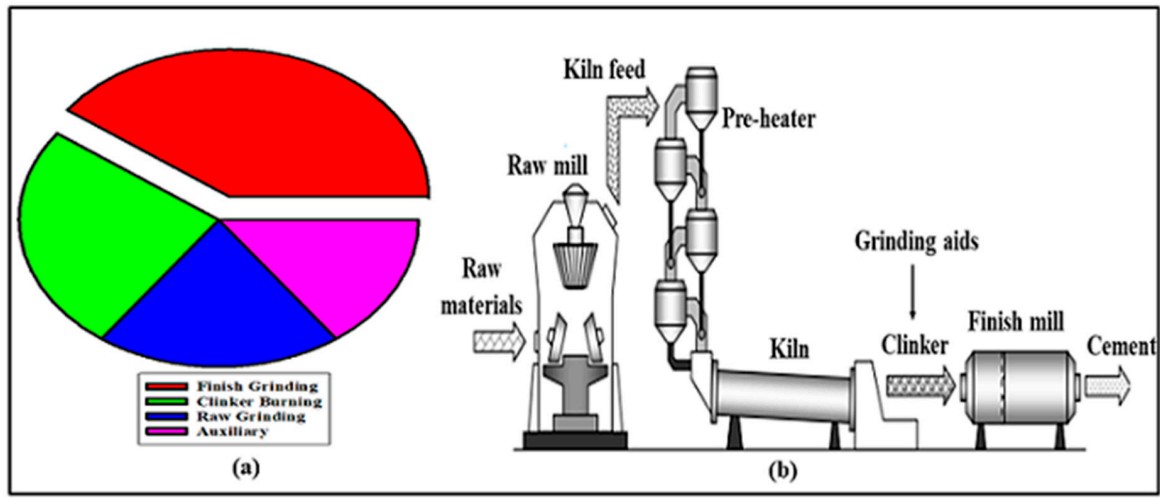

**Figure 10.** (**a**) Energy distribution in cement production equipment (data source: [82]). (**b**) Schematic of dry cement production (modified after [82]).

**Table 6.** Improvement in grindability of cement clinker using grinding aids.

| Grinding Aid | Weight (%) | Improvement in Grinding (%) | Reference |
|---|---|---|---|
| Tri-ethanol amine | 0.1 | 22–29 | [27] |
| Tri-ethanol amine | 0.06 | 16 | [83] |
| Propylene glycol | 0.05 | 25–50 | [27] |
| Organosilicones | 0.05 | 70 | [27] |
| Triisopropanolamine | 0.015 | 26 | [84] |
| Polycarboxylate ether + tri-ethanol amine (1:2) | 0.4 | 22 | [81] |
| Ethylene glycol | 0.05 | 7 | [85] |

Apart from cement clinker fine grinding where chemical additives have shown a beneficial effect on the grindability of material mostly in dry grinding operations, they have also been used for ore comminution in the mining industry. The use of CaO (200 g/t) as an additive for the grinding of magnetite ore (40% Fe, quartz as gangue) has been demonstrated, and has caused the differential grinding of quartz from Fe content [86].

As per the reduction in comminution energy in the wet grinding operation of minerals, sodium silicate, aero801, and sodium oleate have been demonstrated for celestite [87]. At concentrations of 100 g/t and 1000 g/t, sodium silicate and sodium oleate improved the grinding of celestite. At a 10 g/t concentration, both sodium silicate and sodium oleate caused adverse effects (cumulative 80%, passing size is larger than referenced celestite ground at the same operating conditions) on the grinding of celestite at all grinding periods. Aero801 improved the grindability of celestite at 10 g/t, 100 g/t, and 1000 g/t; however, 100 g/t was reported as the appropriate concentration for the grinding of celestite [87].

Downstream Benefits, Economic Assessment, and Industrial Applications of Chemical Additive

Toprack et al. (2014) conducted an industrial scale study to investigate the appropriate chemicals that showed significant improvements in cement production [80]. In the study, six chemicals were investigated for the improvement in production and 28-day strength enhancement of cement (Table 7). The raw materials considered for the cement production were gypsum, limestone, clinker, and fly ash. An existing closed grinding circuit that consisted of a two-compartment ball mill and dynamic air classifier was used for the study. The dynamic air classifier allowed fine particles to move to the silo, while coarse particles returned to the mill for further grinding. The energy consumptions of the reference and the chemically treated samples were compared, as presented on Table 7. In terms of energy saving, all the six chemical additives used in the study reduced the comminution energy significantly. However, the organic and inorganic modified amines, hydroxylamine, and the mixture of polycarboxylate and amines had adverse effects on the 28-day cement strength [80]. An economic analysis of the whole system showed that cement production using any of the chemical additives used in the study was more profitable than cement production without the chemical additive (Table 7).

**Table 7.** Effect of chemical additives on the comminution energy and cement strength enhancement [80].

| Chemical | Dose (g/t) | Energy Saving (%) | Strength Enhancement (%) | Total Cost Saving (Euro/t) |
|---|---|---|---|---|
| Organic and inorganic modified amines | 300 | 14.54 | −3.15 | 0.15 |
| Alkanolamines | 345 | 17.34 | 2.23 | 0.20 |
| Amine acetate (aqueous solution) | 570 | 13.54 | 4.45 | 0.13 |
| Hydroxylamine | 808 | 17.01 | −3.89 | 0.30 |
| Mixture of polycarboxylate and amines | 330 | 16.33 | −4.82 | 0.30 |
| triethanolamine | 331 | 14.37 | 3.53 | 0.24 |

## 4. Electrical Pretreatment

Electrical pretreatment is among the most targeted technologies that has been studied and reported in literature, after microwave pretreatment [21]. Its principle is based on the passage of a high-voltage electrical pulse (HVEP) into the rock matrix to cause fragmentation. The variation in the electrical conductivity of rock causes the expansion and explosion of rock grains when a high voltage is passed into the rock matrix. The non-conductive part of the rock resists the current flow, which leads to a structural change due to HVEP. The expansivity of the mineral grain varies, and therefore micro cracks can be generated in different degrees. A high pressure is also built up within the rock matrix, such that the tensile strength of the rock is exceeded (electrical disintegration (ED) method). This pressure occurs as a result of the change of state (from solid to gas) of some particles within the rock when the electric current passes through the rock lumps [88]. These amount to the deformation and weakening of the rock due to the high temperature (about $10^4$ K) generated by the charge displacement current [89,90]. Different technical terms are found in the literature to represent electrical pretreatment; however, there are slight variations in the procedures or parameters used in creating micro cracks. Some of the technical names are as listed in Table 8.

**Table 8.** Technical terms used for the electrical pretreatment [30].

| Technical Term | Electrode Channel | Voltage Changing Time (ns) |
| --- | --- | --- |
| Electrical disintegration (ED) | Rock | <500 |
| Electrical pulse disaggregation (EPD) | Water | <500 |
| Electrodynamic disintegration (EDD) | Water | <500 |
| Electrohydraulic disintegration (EHD) | Water | >500 |

Electrical pretreatment equipment consists of a high voltage (HV) power source, a sample chamber, and an HV pulse generator that has an arrangement of capacitors with a rectifier. The arrangement of the capacitors depends on the expected capacitance that gives the required voltage. The rock sample to be tested is usually put in water because it has a high dielectric strength and creates a plasma which prevents electrical discharge outside the rock. The rise in voltage is the same for all techniques except electrohydraulic disintegration (EHD), which may result in a lower energy efficiency [30]. For ED, an HV pulse is directly passed into a rock lump that has been immersed in water through the electrode, which makes this procedure quite different from other approaches that require the dipping of the electrode into water in order to generate a shock wave [91]. Electrical pulse disaggregation (EPD), electrodynamic disintegration (EDD), and EHD require water, but more energy is needed for the EHD method and the deformation is generally due to exceeding the compressive strength of the rock [92]. There are divergent opinions on the classification of electrical pretreatment. Some researchers are of the opinion that electrical fragmentation is divided into two categories—one that requires water for breakage and the other without water [93,94]. In this regard, EPD, EDD, and EHD belong to the same group, while ED is the second type. Another view is that it is quite difficult to distinguish between the methods because the electrode gap that is usually associated with EPD, EDD, and EHD may not occur due to rock shape variation [91]. In this case, the classification is based on the voltage rising time.

*Effect of Electrical Pretreatment on Ore Comminution*

The investigation of EPD and ED to be used in mineral processing was started in the early 1970s and research continued until 2002, when an EPD-suitable device (CNT EPD Spark-2) was designed by the research team of CNT Mineral Consulting Inc. (Ottawa, ON, Canada) [95]. The machine has been used to liberate undamaged diamond crystal from the host rock and emerald from quartzite. The good thing about the machine is that the original shape of the crystal is retained, unlike in conventional crushers that can deform the crystal or break it into fine particles. The ED technique was used to disintegrate granite, copper, kimberlite, and nickel sulfide rocks [89]. The feature associated with ED is that the disintegration occurs at the grain boundary without causing unnecessary fine products and liberating valuable minerals [89]. This can reduce the amount of ore to be crushed in a conventional crusher. This method is even more appropriate to be referred to as secondary blasting or pre-crushing, since it is more suitable for larger rock sizes (boulders).

A comparison of ED with a roll-crusher was performed using coal feeds of different specific gravities (1.35–1.45) and size distributions (4.0–5.6 and 5.6–8.0) [94]. The cathode and anode electrodes of the ED device are stainless steel (with a 2 mm sieve size) and brass disks, respectively. The coal samples (Nantun, China) were crushed and sieved to obtain different size distributions, as earlier stated. Representative samples from the two size distributions were mixed (1:1) to get a 200 g feed sample. A total of 100 g of the sample was fed into the sample chamber, such that there existed five layers with 200 g each. The initial voltage supplied through the cathode was 16 kV, and the value was increased up to 56 kV before the sample disintegration occurred. The voltage and current waveforms were studied using the oscilloscope. The ED test was repeated 60 times to arrive at good conclusions. A representative sample prepared as that of the ED test was crushed using a roll-crusher, and a size distribution analysis of both test methods was performed. The results of optical images showed that rough and smooth surfaces were generated for the ED and the roll-crusher, respectively. In addition,

mineral matter was exposed in the case of the ED test products, which indicates that the disintegration occurred at the grain boundaries [94].

The EPD technique was used to liberate minerals from copper (New South Wales, Australia), gold–copper, and lead–zinc (Queensland, Australia) ores [96]. A sample size in the range of 12–45 mm (3600 kg) was collected from mine sites and each ore type was divided into two (one half for the EPD test while the other half was for the conventional crusher test). The products from the two tests were used to carry out a standard bond rod mill test. The closing screen aperture considered for the test was 1.18 mm. The results showed that the percentage changes in the work indices (improvement) between the EPD and conventional crusher for the copper, gold–copper, and lead–zinc ores were 18%, 24%, and 6%, respectively [96]. Similar research was carried out using a platinum group metal ore (South Africa) and samples from Australia, as earlier mentioned. Coarser products were generated in the EPD method, with less fine materials and valuable minerals liberated than that of the conventional crusher [97].

The HVEP technique was used to investigate the liberation of magnetite ore using a −2 mm (200 g) representative feed sample. The sample treated with HVEP and the untreated one were ground under the same grinding conditions using a rod mill. An improvement of 13.19% in the liberation of iron minerals was achieved using HVEP when compared to the untreated sample [98].

Recently, a high-voltage electric pulse crusher (HVEPC) was designed and used for the crushing of phosphate ore [99]. The size fractions of the phosphate ore used in the study ranged between −75 and 50 mm. The bond crushability index of the phosphate ore reduced by 10.6% (compared with the conventional crusher) when the HV pulse-specific energy ranged between 3 and 5 kWh/t. The effects of capacitance, voltage, and PSD of the sample on crushing using the HVEPC were also investigated. It was found that an increase in the capacitance and voltage lead to an improvement in the crushing of the phosphate ore at size ranges of −19 + 12.5, −12.5 + 6.35, and −6.35 + 3.35 mm. The summary of some of the laboratory experiment successes of electrical pretreatment are presented in Table 9.

**Table 9.** Summary of the electrical pretreatment.

| Ore/Mineral | Size (mm) | Electrode Gap (mm) | Voltage (kV) | HVP Specific Energy (kWh/t) | Improvement in Grindability (%) | Reference |
|---|---|---|---|---|---|---|
| Copper | −12.5 | 10–40 | 90–200 | 3 | 18.0 | [96] |
| Gold–copper | −12.5 | 10–40 | 90–200 | 3 | 24.0 | [96] |
| Lead–zinc | −12.5 | 10–40 | 90–200 | 3 | 6.0 | [96] |
| Copper–gold | −12.5 | 10–40 | 90–200 | 3 | 0.0 | [96] |
| Magnetite | −2.0 | 3 | 30 | - | 13.2 | [98] |
| Phosphate | −75 + 50 | - | 40 | 5 | 10.6 | [99] |

Downstream Benefits, Economic Assessment, and Industrial Applications of Electrical Pretreatment

Recently, a pilot scale HV pulses (HVP or EPD) testing machine (Figure 11) developed by SELFRAG AG (Kerzers, Switzerland) was used for the investigation of the particle weakening behavior of ores (gold–copper ore, New South Wales, Australia; iron oxide copper–gold (IOCG) ore, South Australia; and hematite ore) [100]. The machine had a setting system that allowed capacitance and voltage regulation. The pulse energy (ranged 50–200 kV) could be kept constant when varying the voltage or capacitance of the machine. The machine could process ore up to 10 th$^{-1}$ (3–10 th$^{-1}$), depending on the pulse energy, PSD, and density of the ore. The PSDs of the tested ores were 22.4–26.5, 31.5–37.5, and 45–53 mm. It was found that the higher the specific energy of the HVP machine, the better the breakage characteristics measured using the fineness indicator ($t_{10}$), which connotes a cumulative percentage passing size equivalent to one tenth of the original size before the pretreatment and pre-weakening assessments. It was reported that the HVP pretreatment caused a reduction in the competency of gold–copper and IOCG ores by 81.7% and 131.8% respectively, while that of hematite increased by 40.7% [100]. However, an economic evaluation of this method was not performed in the study, which calls for further research. Nevertheless, the HVP machine produced at Julius Kruttschnitt Mineral Research Centre (JKMRC, Indooroopilly, Queensland, Australia) in 2009 consumed considerable energy (1–3 kWh/t) during the ore pretreatment [96]. Safety due to high-voltage generation for rock weakening has usually being

associated with electrical pretreatment; however, this has been put into consideration in the JKMRC machine, and electromagnetic shielding against high electric voltages has been introduced [101]. The electrical method has been applauded for its reduction in the ore to be comminuted; this can assist with rejecting gangue after treatment (pre-concentration) [102,103], and for the reduction in fine particles in the final product after comminution [104]. The former has been suggested to be performed at the mining site so that the haulage cost can be reduced and the rejected can be used as back filling [105]. With that approach, the energy cost can be lowered not only for haulage but also for comminution and processing. The inclusion of the HVP machine in the mining cycle also has environmental benefits, as tailing can be reduced since some wastes would have been rejected from the mining site. A simulation of this approach suggested that 5 kWh/t can be saved for 2000 t/h copper–gold operations [105]. Parker et al. (2015) discussed that electrical pretreatment (electro-comminution) has the potential to improve mineral liberation, which may increase the recovery using the floatation method. In the study, the authors compared the surface chemistry of the untreated and electrical treated samples and found that the latter improved the surface chemistry of the ore as well as the liberation of chalcopyrite in the coarse size range [106].

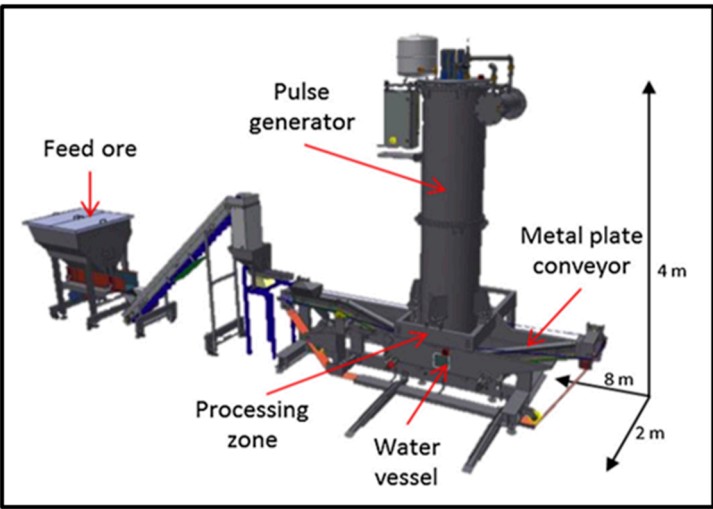

**Figure 11.** High voltage pulse pilot scale testing machine at Kerzers, Switzerland (reproduced with permission) [100].

## 5. Magnetic Pretreatment

Magnetic pretreatment is quite a new method investigated for the reduction in the comminution of iron-rich minerals, especially magnetite. The idea behind the method is that the magnetic pulses generated through a magnetic field may probably change the magnetic dipoles of magnetic minerals and hence introduce microcracks in the rock. This method was proposed in 2016 by J. Yu and his research team [107]. A generator was developed that can create magnetic pulses through a dielectric pipe where coils were mounted. The magnetic field and frequency of the pulses generated were 5.0 T and 0.05 Hz, respectively. The magnetic pulse generator (MPG) was used for the treatment of magnetite ore collected from Dagushan city, Liaoning, China. The representative sample was placed in a plastic container and subjected to magnetic pulse treatment in the MPG for 2.5, 5.0, and 7.5 min using separate samples. The magnetic field intensity of the MPG was kept constant throughout the treatment periods. The pretreated samples and the untreated one (500 g each) were ground for 3 min using a laboratory cone call mill. The energy consumption (measured with a DTZ 119 smart power meter) for grinding untreated and treated samples reflects a 10 W difference, representing 1% energy saving in the grinding process. The PSD analysis performed showed no significant change in the size distribution of products. The improvement in the bond ball mill work index of the treated sample was 0.68% (test sieve size = 74 μm). To sum it up, MPG did not sufficiently weaken the magnetite ore and

there was no significant improvement in the work index. [107]. The magnetic method may provide promising results in future research.

## 6. Ultrasonic Pretreatment

The ultrasonic pretreatment of ore uses a three-dimensional acoustic wave (sound wave above 20 kHz) that passes through the ore matrix to create fractures. The velocity of this wave varies as it passes through the ore media, due to the change in grain size or mineralogy, which causes a kind of reflection and refraction at the grain boundaries. The tensile stress caused by this phenomenon creates microcracks or extends the existing cracks in the ore matrix. The use of ultrasonic waves for the fragmentation of particles was reported to have firstly been attempted by Gärtner in 1953 [52]. In 1981, the Energy and Mineral Research (EMR) company with the support of the US Department of Energy (DOE) constructed an ultrasonic device consisting of a rotating roller that had the capacity to treat 4.5–13.6 kg/h of ore, with an energy requirement of 3 kWh/t for 80% of the product finer than 75 μm. At that time, the energy requirement to achieve the same size as that of using a hammer mill was 20 kWh/t. The major disadvantage of the device, when compared to the conventional machine, was that it had a very low comminution capacity. The DOE, US, in 1988 commenced an investigation into the construction of their own ultrasonic device using the previous idea from 1981. The constructed device was used for the treatment of coal, and resulted in little liberation of the samples. Indeed, the device consumed more energy than that of conventional comminution mills [108]. However, this method has been reported to have improved the grindability of copper ores by up to 32% [108]. Gaete-Garretón et Al. (2000) provided a solution to the limitation (small capacity) of the previous ultrasonic devices by incorporating ultrasonic treatment into a high-pressure roller mill with a capacity of 20 t/day [109]. The developed device and ball mill were used separately for the grinding of prepared copper ore feeds. It was reported that 66% energy was saved when compared with the ball mill. A similar idea was developed called the Ultrasonic High-Pressure Roller Press (UHPRP), which was used in the investigation of the grinding characteristic of copper ore [110]. Findings show that 6% energy was saved when compared with samples without ultrasonic treatment [110]. The combination of microwave and ultrasonic pretreatment was investigated for the disintegration of iron from phosphorus gangue. The findings showed that the disintegration and removal of fine particles from the samples were higher for microwave-treated samples compared to the ultrasonically treated one. The combination of the two methods indicated that a 20% improvement in disintegration was achieved during the process [111]. An ultrasonic device has been piloted for the treatment of carbonate ore which improved its liberation and downstream processes [112]. The economic and industrial applications of this method are limited in the literature; hence, further research is still needed.

## 7. Bio-Milling Pretreatment

Bio-milling is the process of size reduction in particles using living organisms. Since microorganisms can cause the weathering of rock, researchers have therefore suggested that their use for ore comminution could improve the size reduction process. This has found direct application in the milling of particulate matter in the nano range. The use of fungal biomass for milling chemically synthesized $BiMnO_3$ had been reported [113]. It was discussed that $BiMnO_3$ (150–200 nm) was reduced to <10 nm without any effect on either the crystallinity or phases of the material [113]. Research using this approach for ore comminution energy reduction may find relevance in the future.

## 8. Ranking of Methods of Ore Pretreatment

To look for an immediate way forward to reduce comminution energy, Canada Mining Innovation Council (CMIC) ranked both conventional and some emerging technologies based on the responses and analysis of a questionnaire filled in by professionals in the mining sector [114]. Consideration was given to energy reduction, costs, safety, stage of the technology, and downstream benefits. For all these parameters considered for rating the pretreatment methods, 1–5 were used as rating numbers

in an ascending order of benefit, except cost. For the cost implication of the methods, the rating five means that the cost is very low, meaning that one is the costliest rating number [114]. Based on the available literature [80,100,107], there are developments in the stages of technologies, which was part of the ranking matrix used by the CMIC; hence, there is a need for updating the ranking of emerging technologies so that research can be channeled toward the proven methods (Table 10).

**Table 10.** Ranking of ore pretreatment methods (* cement production) [114].

| Methods | Energy Reduction | Cost | Safety | Scale of Application | Downstream Benefits | Total |
|---|---|---|---|---|---|---|
| Conventional heating | 2 | 1 | 2 | 2 | 2 | 9 |
| Microwave | 3 | 2 | 3 | 4 | 3 | 15 |
| Radiofrequency | - | 2 | 3 | 1 | 1 | 7 |
| Chemical * | 2 | 3 | 3 | 4 | 4 | 16 |
| Electrical | 3 | 2 | 3 | 4 | 4 | 16 |
| Magnetic | - | - | 3 | 1 | - | 4 |
| Ultrasonic | 1 | 1 | 2 | 3 | 3 | 10 |
| Bio-milling | - | - | 2 | 1 | - | 3 |

## 9. Conclusions

Comminution is an important operation to liberate minerals/ores. It accounts for about 50% to 70% of the total electrical energy required in mining activities. Many methods have been explored to reduce this energy demand. These methods include thermal (via furnace, microwave, and radiofrequency techniques), chemical, electrical, magnetic, ultrasonic, and bio-milling. Thermal pretreatment via furnace showed that improvements in the grindability of some ore can be achieved by up to 45%. Nevertheless, the high energy demand of furnaces, their non-uniform heating, safety issues, and environmental pollution due to their release of process gasses are major concerns. The development of a solar convergence device that can produce high thermal energy to heat ore can be investigated, since solar energy is environmentally friendly and may be cheaper in the long run. Microwave dielectric heating is the most pursued of all methods, with promising results at both laboratory and pilot scale studies. Findings from the former suggested that a 3–92% improvement in the grindability of some ores can be achieved, while the latter shows a maximum of 9% reduction in the grindability of copper ore. In contrast to the microwave, radiofrequency dielectric heating has not produced promising results for some selected rocks. The chemical method has been demonstrated to be appropriate to aid comminution, especially in the cement industry, with an improvement in the grindability of clinker in the range of 7–70%. However, the cost of the chemicals used in the process is still a challenge. The results of studies using electrical and ultrasonic methods showed improvements in grindability of up to 24% and 66%, respectively. The former has been piloted for gold, copper, and iron-related ores, while the latter has been used for carbonate rocks with promising results. For magnetic and bio-milling methods, progress has not been made, as suitable approaches for ore comminution energy reduction have not been found, but these methods may find relevance in the future. To sum up, microwave and electrical pretreatments should be given preferential attention based on their stage of technology, energy reduction, cost, safety, and downstream benefits.

**Author Contributions:** Conceptualization, S.O.A. and H.A.M.A.; methodology, S.O.A.; formal analysis, S.O.A.; investigation, S.O.A.; resources, S.O.A., H.A.M.A.; data curation, S.O.A.; writing—original draft preparation, S.O.A.; writing—review and editing, S.O.A., H.A.M.A.; visualization, H.A.M.A, H.M.A.A.; supervision, H.A.M.A, H.M.A.A.; funding acquisition, H.M.A.A., H.A.M.A. All authors have read and agreed to the published version of the manuscript.

**Funding:** This project was supported by the Deanship of Scientific Research (DSR), King Abdulaziz University, Jeddah under grant No. (DG-026-306-1441). The authors, therefore, gratefully acknowledge the DSR technical and financial support.

**Acknowledgments:** The authors also acknowledge the contributions of anonymous reviewers that improved this work.

**Conflicts of Interest:** The authors declare no conflict of interest.

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
