# Peer review of "Methods of Ore Pretreatment for Comminution Energy Reduction"

_minerals, doi:10.3390/min10050423_

Round 1

Reviewer 1 Report

The authors make a comprehensive review on the pretreatment methods applied to reduce the energy consumed for comminution. The authors make a good literature reviews on the pretreatment methods include thermal (via oven, microwave or radiofrequency), chemical additive, electric, magnetic, ultrasonic, and bio-milling. The authors concluded that microwave and electrical pretreatments showed positive results. Overall, I recommend this manuscript for publication after minor revision.

The authors concluded that microwave is a promising technique and showed positive results. But the authors should mention the energy consumption of microwave heating, which is the major drawback of this process. What is the economic feasibility of using microwave pretreatment?

The author should mention the remarkable works of Omran et al. on the microwave assisted comminution on the thermal pretreatment section:

- Omran M, Fabritius T & Mattila R (2015) Thermally assisted liberation of high phosphorus oolitic iron ore: A comparison between microwave and conventional furnaces. Powder Technology, 269: 7–14.

- Omran M, Fabritius T, Abdel-Khalek N, El-Aref M, Elmanawi AE.-H, Nasr M &Elmahdy A (2014) Microwave assisted liberation of high phosphorus oolitic iron ore. J Miner Mater Charact Eng 2: 414-427.

- Omran M, Fabritius T, Elmahdy A, Abdel-Khalek N & Gornostayev S (2015) Improvement of phosphorus removal from iron ore using combined microwave pretreatment and ultrasonic treatment. Separation and Purification Technology, 156 (2015) 724–737

Author Response

Thank you for efforts and contributions to this manuscript.

  1. Economic feasibility of microwave pretreatment: The economic assessment of microwave pretreatment that considered energy consumed by microwave into consideration has been added to the manuscript as requested. It can be found between line 325 -368.
  2. This paper “Omran M, Fabritius T & Mattila R (2015) Thermally assisted liberation of high phosphorus oolitic iron ore: A comparison between microwave and conventional furnaces. Powder Technology, 269: 7–14) has been reviewed and added to the manuscript.” It can be found between lines 221 – 231 (Reference number 36).
  3. This paper “Omran M, Fabritius T, Abdel-Khalek N, El-Aref M, Elmanawi A E.-H, Nasr M Elmahdy A (2014) Microwave assisted liberation of high phosphorus oolitic iron ore. J Miner Mater Charact Eng 2: 414-427” has been reviewed and added to the manuscript. It can be found between line 206 - 211 (Reference number 57).
  4. This paper “Omran M, Fabritius T, Elmahdy A, Abdel-Khalek N & Gornostayev S (2015) Improvement of phosphorus removal from iron ore using combined microwave pretreatment and ultrasonic treatment. Separation and Purification Technology, 156 (2015) 724–737.” It can be found between line 594 – 598 (Reference number 114).

Reviewer 2 Report

Please find attached the comments.

Author Response

Thank you for contributions to this manuscript.

  1. The book has been added as suggested “Austin, A.G.; Luckie, P.T.; Klimpel, R.R. Process Engineering of Size Reduction: Ball Milling; Society of Mining Engineers of the American Institute of Mining, Metallurgical and Petroleum Engineers: new York, 1984; ISBN 0895204215.” It caan be found between line 37 – 40 (Reference number 10).
  2. The discussion of energy consumption in some countries has been added. The trend of a of comminution energy of a selected country (Canada; as a case study) was added and can be found between lines 30 - 55.
  3. The paper “Toprak et al. The influences and selection of grinding chemicals in cement grinding circuits, Construction and Building Materials 68 (2014) 199–205.” Which elaborately discussed the industrial application of the Grinding aids was reviewed and added as suggested. It can be found between line 421 - 437.
  4. The methods that have been either used at pilot or industrial scales have been added as suggested (lines 139 – 163 (Thermal via furnace), 282 – 368 (Microwave pretreatment), 421 – 437 (Chemical additives), and 517 – 552 (Electrical pretreatment)).
  5. The processing methods used for the downstream benefits of the pretreatment methods are discussed together with the industrial uses (lines 139 – 163 (Thermal via furnace), 282 – 368 (Microwave pretreatment), 421 – 437 (Chemical additives), and 517 – 552 (Electrical pretreatment)).
  6. The improvement in grade and recovery and other downstream benefits are discussed together with the industrial uses. Table 10 on line 621 was drafted from the discussion of the methods (downstream benefits, economic feasibility (for those that have been performed according to the available literature) and stage of the technology; laboratory, pilot or industrial scale). The report of Canada Mining Innovation Council (CMIC), which was the available document that rated the present comminution practice in mining industry and emerging technologies was used as a reference guide to form the table.
  7. The economic feasibility of thermal pretreatment via furnace, microwave, chemical and electrical pretreatment have been added (discussed together with downstream benefits and industrial applications). (Lines 139 – 163 (Thermal via furnace), 282 – 368 (Microwave pretreatment), 421 – 437 (Chemical additives), and 517 – 552 (Electrical pretreatment)).
  8. Thank you once again. Your suggestions have improved the manuscript.

Reviewer 3 Report

This is a nice comprehensive paper.   I think there should be included more judgement comments and opinions.  

There is possible an important face to include at the relevant point in the paper about the intrinsic hardness value of a mineral or ore particle, pretreated or otherwise.    As a particle decreases in size, it increases in hardness (grossly speaking).  Therefore there must be a point at which any kind of pre-treatment has no effect on the particles.  What that particle size (shape or texture ), is I don't believe has been universally identified by these methods.  That would need some further investigation.

Microwave/dielectric heating does not have the same effect on all minerals in the same way, nor ores.  Texture, crystalinity, sulphides, carbonates and hydrated all heat differently in the field.  There is a thesis at the University of Nottingham (Selective Heating of Ores and Minerals, by Louisa Groves 2008) which could provide further information on this - note sure if there is an electronic copy, you'd have to go through Professor Sam Kingman.)

Line 62 - Yes, but reality suggests this isn't the case.  Grains aren't 'rearranged'. You need to address what you mean.  If you mean displaced due to the creation of fracture, then this would be more appropriate.

Microwave heating, when applied is just that, microwave heating, but with some ores, they can heat without 'reaction' and then heat can transfer through the sample, resembling a kind of conventional heating transfer. 

Some minerals require a lot of applied power and residence time before showing heating or comminution-type effects.  Example: Goethite - hydrated iron ore.  The iron mineralogy heats, but the bulk ore must heat to a point to where the -OH (water) is released.  

Line 73 - Make sure you are clear that it is granites being described, as per the reference.

Line 75 - ergo, would a euhedral crystal (ore / rock texture) respond better than an amorphous mineral?  So ore or rock texture, the physio-chemical profile are key to the success of any pre-conditioning.

Table 1 - Maybe specify 'mineral' and not ore.  Albeit quartzite is a metamorphic rock,. it comprises a single mineralogy - quartz.

Question:  Why do you think that gypsum shows an increase in hardness?  It's incredibly soft (Moh's scale 1).  Do you think it reacts similar to that of goethite i.e. the water is driven from the mineralogy? CaSO42H2O....  A few comments on this would be ideal.  Cassiterite - a lot of heat to create a reaction.  Apart from an energy balance versus economic value - which was covered later on in your text - do you think, the crystal shape  of the minerals has an effect on their ability to fracture either either intergranularly or intragranularly?  Maybe some comments would go well there.

The same questions apply to samples which would have microwave pretreatment, not just furnace (conventional heating)

Table 2 - Ore/Mineral, it's not just ore in your column heading.

Table 6 - not really understanding the significance - can it be integrated in the Table 7?

Line 404  "liberate targeted ore from gangue"  It's targeting responsive minerals from non-responsive minerals.  Gangue minerals can respond during treatments, but which ones?  MW treatment discriminate transparent minerals to minerals of interest. Same with Selfrag and chemical.  Conventional is somewhat different at time.

This is a good paper, but I would love to see more insight complementing the review comments.   There are minor changes I've suggested, it's the insight I think needs to be worked upon a little more.

Author Response

Thank you for your comments and suggestions.

  1. The hardness of mineral before and after microwave pretreatment has been added. The size at which microwave pretreatment has no improvement in minerals’ weakness are limited in literature, hence, needs for further studies (as you suggested) (line 222 - 254).
  2. I contacted Prof. S. Kingman for the suggested thesis, however, electronic copy was not available. Nevertheless, other available literature was used to discuss the ore response to microwave treatment. Also, effect of ore size, crystallinity, grain size, and texture on ore response to microwave treatment have been added (all can be found within line 222 - 257).
  3. Line 62 (now line 77 - 79). The sentence has been reframed as suggested.
  4. A situation where microwave dielectric heating is like the conventional heating (via furnace) has been discussed as suggested (line 188 - 192).
  5. Line 73 (now lines 89 - 91); the issue has been addressed as requested.
  6. Line 75 (now line 89 -92). A sentence has been added in order not to generalize the ore response to pre-conditioning.
  7. Table 1; line 138 (Adjusted as required).
  8. Gypsum being one of the hydrate minerals and can decompose easily due to removal of water molecules when exposed to temperature within 100 – 250 o When dehydration occurs, gypsum (CaSO4.2H2O) may transformed to plaster of Paris (hemihydrate mineral; CaSO4.O.5H2O) which usually lead to structural failure (116 -123). Discussion of crystal shapes have been introduced (Line 91 -94 and line 199 - 201).
  9. Table 2 (addressed as suggested).
  10. Table 6 has been integrated into Table 10 (former table 7).
  11. Line 404 (now line 623); the correction has been made. Target mineral depends on its usefulness whether responsive or non-responsive to pre-conditioning.

Round 2

Reviewer 2 Report

The reviewer suggests "accept" of the manuscript. The authors adressed the points raised by the reviewer.

Reviewer 3 Report

No further comments.